# Genetic Analyses of Rabbit Survival and Individual Birth Weight

**DOI:** 10.3390/ani12192695

**Published:** 2022-10-07

**Authors:** Rafik Belabbas, Rym Ezzeroug, Ali Berbar, María de la Luz Garcia, Ghania Zitouni, Djamel Taalaziza, Zoulikha Boudjella, Nassima Boudahdir, Samir Diss, María-José Argente

**Affiliations:** 1Laboratory of Biotechnologies Related to Animal Reproduction, Institute of Veterinary Sciences, University Blida, B.P 270, Road of Soumaa, Blida 09000, Algeria; 2Centro de Investigación e Innovación Agroalimentaria y Agroambiental (CIAGRO-UMH), Universidad Miguel Hernández de Elche, Ctra. Beniel km 3.2, 03312 Alicante, Spain; 3Technical Institute of Animal Breeding, Bab Ali, Alger 16111, Algeria

**Keywords:** birth weight, kit survival, genetic parameters, threshold model, rabbits

## Abstract

**Simple Summary:**

Kit survival in the first hours after farrowing has been related to the birth weight of kits. In prolific species, newborn survival is controlled both by the genes of the newborns that are involved in vitality, health, and growth (direct genetic effects), and by the dam effects that affect milk yield and other mothering abilities (maternal effects). Genetic parameters of peri and postnatal survival have been estimated traditionally on the performance of dam (assuming normally distributed continuous traits), but it is more appropriate to consider as categorical traits of the kit. The objective of this study was to estimate the heritabilities of kit survival at birth and weaning, as well as the individual birth weight, and the genetic correlations between those survival traits and birth weight using a combined linear threshold model with a Bayesian approach. Heritabilities of survival at birth and weaning, as well as birth weight, were low (0.021 and 0.027) for survival traits and slightly greater (0.146) for birth weight after adjusted litter size. No genetic correlation was found between survival traits. Genetic correlation between survival at birth and birth weight showed a positive value (+0.134 and +0.535 after being adjusted for litter size). These magnitudes of genetic parameter estimates suggested that there is substantial potential for the genetic improvement of kit survival at birth through selection for birth weight.

**Abstract:**

Genetic parameters of kit survival traits and birth weight were estimated on ITELV2006 synthetic line aimed at improving kit survival using a multiple trait linear and threshold model. Data on 1696 kits for survival at birth and at weaning, as well as individual birth weight and litter size were analysed. Genetic effects of kit survival traits and birth weight were estimated based on threshold and Gaussian models, respectively, using a Bayesian approach. The statistical model included, as fixed effects, parity, lactation status, season of farrowing, nest status, cannibalism in kit, place of kit’s birth in the cage and gender, and adjustment for litter size. Posterior means of heritabilities for direct genetic effects of survival at birth and the entire nursing period, as well as birth weight, were 0.018, 0.023, and 0.088, respectively, and were increased when adjusted for litter size to 0.021, 0.027 and 0.146. Genetic correlation between survival traits was zero. Therefore, these traits can be treated genetically as different traits. Genetic correlation between direct effects of survival at birth and birth weight showed positive, but low, value (+0.134) and was increased to +0.535 when the traits were adjusted for litter size. No genetic correlation was found between survival at weaning and birth weight. These magnitudes of genetic parameter estimates suggested that there is substantial potential for the genetic improvement of kit survival at birth through selection for birth weight.

## 1. Introduction

The number of weaned kits per dam is the main driver of the profitability of rabbit meat production [1]. Therefore, breeding programs have focused on the genetic improvement of litter size, leading to a substantial increase in the total number of kits born [2]. However, an increase in the total number of kits born leads to a lower birth weight and increased kit mortality [3]. The low birth weight of kits has been shown to be influenced by the available uterine space per fetus and on the blood supply [4,5], which could result in physiological immaturity and failure to maintain the body temperature that is crucial for kit survival in the first hours after farrowing [3].

In prolific species such as pigs and rabbits, newborn survival is controlled both by the genes of the newborn that are involved in vitality, health, and growth (direct genetic effects), and by the dam effects that affect milk yield and other mothering abilities (maternal effects), which is a challenge for improving newborn survival genetically [6]. Genetic parameters of peri and postnatal survival have been estimated traditionally on the performance of dam (assuming normally distributed continuous traits) by fitting a linear model using restricted maximum likelihood (REML) [7,8,9,10]. However, newborn survival is regarded as a categorical trait of offpring that is more appropriately analysed using a threshold model in order to estimate direct genetic effects of newborn survival and correlation to maternal effects [11]. The genetic determination of peri- and postnatal survival must be studied before including it as a selection objective in the rabbit breeding. To our knowledge, there is no information in rabbits about the direct heritabilities for kit survival traits using a threshold model and the genetic correlations between birth weight and kit survival. The objective of this study was to estimate the heritabilities of kit survival at birth and at weaning, as well as the individual birth weight, and the genetic correlations between those survival traits and the individual birth weight using a combined threshold-linear model with a Bayesian approach.

## 2. Materials and Methods

### 2.1. Ethics Statement

This study was approved by the scientific council of the Biotechnology Laboratory of Animal Reproduction, part of the Institute of Veterinary Sciences at the University of Saad Dahleb Blida (Blida, Algeria).

### 2.2. Animals, Housing and Feeding

The animals came from the ITELV2006 synthetic line. This line was created as part of a co-operative rabbit project between the Institut Technique des Elevages (ITELV, Algeria) and Institut National de la Recherche Agronomique (INRA, France) by means of inseminating does from a local Algerian rabbit population with the semen of bucks from the INRA2666 synthetic line [12]. The ITELV2006 synthetic line has been maintained in discrete generation without selection and by avoiding inbreeding since its foundation.

Does and bucks were individually housed in wired flat-deck cages in a building equipped with a cooling system. They were kept under a consistent photoperiod of 16L:8D (light:dark hours). Animals were fed a standard commercial pelleted diet ab libitum (16% crude protein, 15% crude fibre and 2.6% ether extract) and had free access to water. Natural mating was performed. Adult males (7–10 months of age) were used. A maximum of two matings per week were carried out, and the same feeding protocol was used throughout the experiment, in order to qualitatively and quantitatively guarantee the quality of the semen. Does were mated first at 20 weeks of age and at 10 d after parturition. If the does refused to be mated, they were again mated a week later. This implies that some of the does carried out gestation and lactation at the same time.

Three days before parturition, nests were placed into the cages. Every morning, after 8 am, all the nest boxes were revised, and status of the nest, presence of cannibalism in kits, and if kits were born inside or outside of the nest was recorded. Assessment of nest quality was based on presence of doe’s hair in the nest (bad, there was no hair in the nest because the female did not prepare it for delivery; intermediate, >50% of the nest had material covered with hair; excellent, only hair was observed) as described by Blumetto et al. [13]. Kits born in the first three parities were individually weighed, sexed, and identified within 24 hours after birth. Adoptions were not practiced. Weaning took place at 35 d of age. The study was conducted from June to November 2017. Table 1 shows the temperature and relative humidity by month. Summer runs from June 1 to August 31, and fall from September 1 to November 30 at the location of the study. A total of 1696 kits from 208 litters of 81 does were weighed individually at birth using a digital scale that could measure weights up to 500 g with a precision of 0.01 g. The pedigree file included 1920 individuals.

### 2.3. Genetic Analysis

The traits analysed were survival at birth (SB), survival at weaning (SW), and individual birth weight including dead kits (BW). In the genetic analyses, survival traits were coded as one (dead) and two (alive); zero was interpreted as a missing value. Dead kits at birth were treated as missing observations in the trait survival at weaning. Multiple trait Bayesian analyses were carried out using a threshold model for survival traits and a linear Gaussian model for birth weight. The multiple trait animal model is as follows:**y** = **Xb** + **Za** + **Wm** + **Vc** + **e**(1)
where y includes the unobservable underlying continuous variable (liability) for survival traits and the observed phenotypic observations of birth weight of each individual kit. The underlying continuous liability was linked to the observed binary observation of kit survival through a threshold. Vector **b** included the systematic effects of parity (1st, 2nd and 3rd parity), lactation status (lactating and non-lactating doe at mating), season of farrowing (summer and autumn), nest quality (bad, intermediate and excellent), cannibalism in kit (yes or not), born inside of the nest (yes or not) and gender of kit. The vectors **a**, **m**, **c** and **e** represent the direct additive genetic effects of the kits, maternal permanent environmental and common litter effects of the does and the environmental residual effects, respectively. **X**, **Z**, **W** and **V** are incidence matrixes linking the effects with y. In further analyses, survival at birth and weaning and individual birth weight were additionally adjusted for number of kits born in order to to assess the influence of litter size on the estimates.

The variance–covariance structure was:(2)$$V\;\left[\begin{array}{c} \begin{array}{c} a\\ m\\ c \end{array}\\ e \end{array}\right]=\left[\begin{array}{c} \begin{array}{c} \begin{array}{cc} A\;⊗\;G_{0} & \begin{array}{ccc} 0 & 0 & 0 \end{array} \end{array}\\ \begin{array}{cc} 0 & \begin{array}{ccc} I\;⊗\;M_{0} & 0 & 0 \end{array} \end{array} \end{array}\\ \begin{array}{cc} 0 & \begin{array}{ccc} 0 & I\;⊗\;C_{0} & 0 \end{array} \end{array}\\ \begin{array}{cc} 0 & \begin{array}{ccc} 0 & 0 & I\;⊗\;R_{0} \end{array} \end{array} \end{array}\right]$$
where **A** and **I** are the additive genetic relationship matrix and identity matrix, respectively. **G**_0_ represents the variance and covariance matrix of the direct additive genetic effects of the kits and has the following structure:(3)$$G_{0}=\left[\begin{array}{ccc} \sigma_{\mathrm{aSB}}^{2} & \sigma_{a\;\mathrm{SB}\;\mathrm{SW}\;} & \sigma_{a\;\mathrm{SB}\;\mathrm{BW}}\\ \sigma_{a\;\mathrm{SB}\;\mathrm{SW}} & \sigma_{\mathrm{aSW}}^{2} & \sigma_{a\;\mathrm{SW}\;\mathrm{BW}}\\ \sigma_{a\;\mathrm{SB}\;\mathrm{BW}} & \sigma_{a\;\mathrm{SW}\;\mathrm{BW}} & \sigma_{\mathrm{aBW}}^{2} \end{array}\right]$$

**M**_0_ represents the variance and covariance matrix of the maternal permanent environmental effects and has the following structure:(4)$$M_{0}=\left[\begin{array}{ccc} \sigma_{\mathrm{mSB}}^{2} & \sigma_{m\;\mathrm{SB}\;\mathrm{SW}\;} & \sigma_{m\;\mathrm{SB}\;\mathrm{BW}}\\ \sigma_{m\;\mathrm{SB}\;\mathrm{SW}} & \sigma_{m\;\mathrm{SW}}^{2} & \sigma_{m\;\mathrm{SW}\;\mathrm{BW}}\\ \sigma_{m\;\mathrm{SB}\;\mathrm{BW}} & \sigma_{m\;\mathrm{SW}\;\mathrm{BW}} & \sigma_{m\;\mathrm{BW}}^{2} \end{array}\right]$$

**C**_0_ represents the variance and covariance matrix of the common litter effects and has the following structure:(5)$$C_{0}=\left[\begin{array}{ccc} \sigma_{c\;\mathrm{SB}}^{2} & \sigma_{c\;\mathrm{SB}\;\mathrm{SW}\;} & \sigma_{c\;\mathrm{SB}\;\mathrm{BW}}\\ \sigma_{c\;\mathrm{SB}\;\mathrm{SW}} & \sigma_{c\;\mathrm{SW}}^{2} & \sigma_{c\;\mathrm{SW}\;\mathrm{BW}}\\ \sigma_{c\;\mathrm{SB}\;\mathrm{BW}} & \sigma_{c\;\mathrm{SW}\;\mathrm{BW}} & \sigma_{c\;\mathrm{BW}}^{2} \end{array}\right]$$

**R**_0_ represents the variance and covariance matrix of the residual environmental effects and has the following structure:(6)$$R_{0}=\left[\begin{array}{ccc} \sigma_{e\;\mathrm{SB}}^{2} & \sigma_{e\;\mathrm{SB}\;\mathrm{SW}\;} & \sigma_{e\;\mathrm{SB}\;\mathrm{BW}}\\ \sigma_{e\;\mathrm{SB}\;\mathrm{SW}} & \sigma_{e\;\mathrm{SW}}^{2} & \sigma_{e\;\mathrm{SW}\;\mathrm{BW}}\\ \sigma_{e\;\mathrm{SB}\;\mathrm{BW}} & \sigma_{e\;\mathrm{SW}\;\mathrm{BW}} & \sigma_{e\;\mathrm{BW}}^{2} \end{array}\right]$$

Bayesian analyses were carried out using Gibbs sampling in order to estimate the variance components of survival traits and individual birth weight. In these analyses, bounded uniform priors were used for the systematic effects with:b α constant.(7)

The conditional prior distributions for the additive genetic, maternal litter and residual environmental effects were sampled from a multivariate normal (N) distribution with(8)$$a|A,\;G_{0}\sim N(0,\;A\;⊗\;G_{0})$$



(9)
$$m|I,\;M_{0}\sim N(0,\;I\;⊗\;M_{0})$$





(10)
$$c|I,\;C_{0}\sim N(0,\;I\;⊗\;C_{0})$$


(11)
$$e|I,\;R_{0}\sim N(0,\;I\;⊗\;R_{0}),\;\mathrm{respectively}$$



Statistical inferences were derived from samples of the marginal posterior distribution obtained by Gibbs sampling as implemented in the program TM [14]. The heritabilities and genetic correlations of the traits were calculated as posterior means from the marginal distributions of these parameters. We used a chain of 1,000,000 samples, and burn-in of 500,000; only 1 out of every 100 samples was saved for inferences. Convergence was tested using the Z criterion of Geweke [15], and Monte Carlo sampling error was computed using time-series procedures, as described in [16]. The Monte Carlo standard errors were small, and lack of convergence was not detected by the Geweke test. In order to identify the precision of the parameters, the 95% highest posterior density (HPD) intervals were determined from their marginal posterior distributions.

## 3. Results

### 3.1. Descriptive Statistics

The mean birth weight, including dead kits, was 1.43 g lower and 1.05 g higher standard deviation than those excluding weights of dead kits. Losses occurring at birth were similar to during the entire nursing period (9.44% and 10.59%, Table 2). The weight of kits born is affected by environmental factors such as parity, lactation status, season of farrowing, nest quality, cannibalism in kit, place of kit’s birth in the cage and sex. For example, this study reported a lower birth weight in live and dead kits from nulliparous females vs. multiparous ones, from lactating females vs. non-lactating ones, from summer vs. autumn, from cannibalism vs. non-cannibalism, from inside vs. outside of nest, and from females vs. males (Table 3). Survival of kits at weaning seems to be related to individual birth weight. In this regard, we found that rabbits that did not survive to weaning weighed between 10 and 30% less at birth than those that survived (Table 3).

### 3.2. Heritabilities

The posterior means of heritabilities for kit survival were 0.018 at birth and 0.023 at weaning, whereas the corresponding estimate for individual birth weight was higher (0.088) compared with survival traits (Table 4). These heritabilities were significantly different from zero as indicated by their 95% HPD intervals in the range frorm 0.001 to 0.052 and 0.006 to 0.234, respectively. Slightly higher heritabilities were obtained after adjustment for litter size at birth (0.021, 0.017 and 0.146 for survival at birth and at weaning, and individual weight at birth, respectively, Table 5).

### 3.3. Genetic Correlations

Genetic correlation between direct genetic effects of survival at birth and at weaning was small (−0.111, Table 4) and not different from zero as indicated by its symmetric HPD at 95% that included zero (ranged from -0.763 to +0.613, Table 4) and the low probability to be lesser than zero (P = 0.67, Table 4). Direct genetic effects of birth weight showed a slight positive association with survival traits (+0.134 for survival at birth and +0.041 for survival at weaning), but the probability of being greater than zero was low (0.63 and 0.47, respectively, Table 4).

Apart from the correlation between survival at birth and at weaning, adjustment for litter size resulted in an increase in correlation among birth weight and survival traits (+0.535 for survival at birth and +0.083 for survival at weaning, Table 5). However, the probability of the genetic correlation being greater than zero was only high between birth weight and survival at birth (P = 0.87, Table 5). In particular, we can see that the posterior distribution of correlation between birth weight and survival at birth was negatively skewed and resulted in a large HPD interval (Figure 1).

### 3.4. Maternal Permanent Environmental and Common Litter Effects

The phenotypic fractions of the maternal permanent environmental variance for survival traits and birth weight (0.024 to 0.070, Table 6) were similar to phenotypic fraction of the direct genetic variances (i.e., heritabilities) of the same traits (Table 4). Note that these effects include both maternal non genetic and genetic effects. Except for the fact that the correlation between maternal permanent environmental effects of birth weight and survival at birth showed a larger negative value (−0.437, P = 0.85, Table 6), the rest of the correlations between the maternal permanent environmental effects of survival traits and birth weight were of similar magnitude to the corresponding correlation between the direct genetic effects. Adjustment for litter size decreased the phenotypic fractions of the maternal permanent environmental variance in particular of birth weight. Accordingly, the correlations among maternal permanent environmental effects of birth weight and survival traits were increased (−0.437 vs. −0.702 with P = 0.95 for survival at birth and +0.098 vs. −0.273 for survival at weaning with P = 0.68, Table 7).

The phenotypic fractions of the variances common to all kits within a litter were 0.199 for survival at birth, 0.234 for survival at weaning and 0.435 for birth weight (Table 8), substantially higher than the phenotypic fraction of the direct genetic variances and maternal permanent variances of the same traits. Furthermore, correlations among litter effects of birth weight with those of survival traits were of higher magnitude than corresponding correlations among direct genetic effects (+0.134 vs. +0.633, P = 0.99 for survival at birth and +0.041 vs. +0.362 for survival at weaning, P = 0.99 see Table 8). Adjustment for litter size decreased by half the phenotypic fraction of the litter variance of birth weight, but correlations among litter effects of birth weight with survival traits were not affected (Table 9).

## 4. Discussion

The present study shows that 20% of kits born perished before weaning, which is in the range values of those published in maternal rabbit lines by García and Baselga [9] and Badawy et al. [10]. Preweaning losses were estimated to be 1–2% at birth, 70–78% in the first week and 17% in the second week [17,18,19]. Decreased kit survivability raises animal welfare concerns and limits successful rabbit meat production.

Success of selection for improved kit survival depends on that trait having variability and being heritable. Kit survival traits presented moderated variability in this study, e.g., coefficients of variation ranged from 25 to 27%. In relation to heritability, it should be noted that genetic parameters for newborn survival traits in prolific species such as pigs and rabbits have mainly been estimated by treating survival as a trait of the dam using, for example, number of stillborn or its proportion of the number born [7,8,9,10]. In the present work, kit survival was regarded as a trait of the kit in order to estimate direct genetic effects of kit survival. Our estimates of direct heritabilities for survival at birth and at weaning showed low and non-relevant values, in agreement with values reported in pigs when survival traits were ascribed to piglets, with those estimates varying from 0.01 to 0.18 [6,20,21,22,23,24,25,26]. To our knowledge, there is no information about direct heritabilities for survival traits analysed using a threshold model in rabbits, but estimated heritabilities for those traits considered as dam traits were also low and non-relevant, varying from 0.03 to 0.07 [10]. For birth weight, the estimated heritabilities in this study were within the range of 0.02–0.26 reported in pigs [6,20,22,23] and in rabbits [27].

The genetic correlations between direct genetic effects of survival traits were near zero. Consequently, these peri and postnatal survival traits can be treated genetically as different traits. Our results agree with the low genetic correlation between direct effects of piglet survival at birth and during the nursing period reported by Roehe et al. [6], Lund et al. [21] and Su et al. [23]. In relation to genetic correlations between birth weight and survival at birth, our estimates showed values between 0.134 and 0.535, after adjustment for by litter size. However, birth weight was not related to survival at weaning. Similar values have been reported in swine [6,23]. It should be noted that one difficulty in estimation of genetic correlations between stillbirths and birth weight may be nonlinear genetic associations between these traits [25,28,29]. The methodology used in the present study assumes linear associations between those traits. However, the association between birth weight and survival could be nonlinear. Generally, selection for birth weight should be to an optimal birth weight in order to avoid a potential increase in stillbirth due to heavy kits, as occurred in piglets.

## 5. Conclusions

The results of the present study showed that kit survival at birth and during the nursing period is under genetic control, but the low heritabilities would limit the success of selection. Birth weight displayed a positive genetic correlation with kit survival at birth and a higher heritability than survival traits. Therefore, selection for birth weight could be a useful tool for the improvement of kit survival at birth. However, further studies on this species are required to validate our results.

## Figures and Tables

**Figure 1 animals-12-02695-f001:**
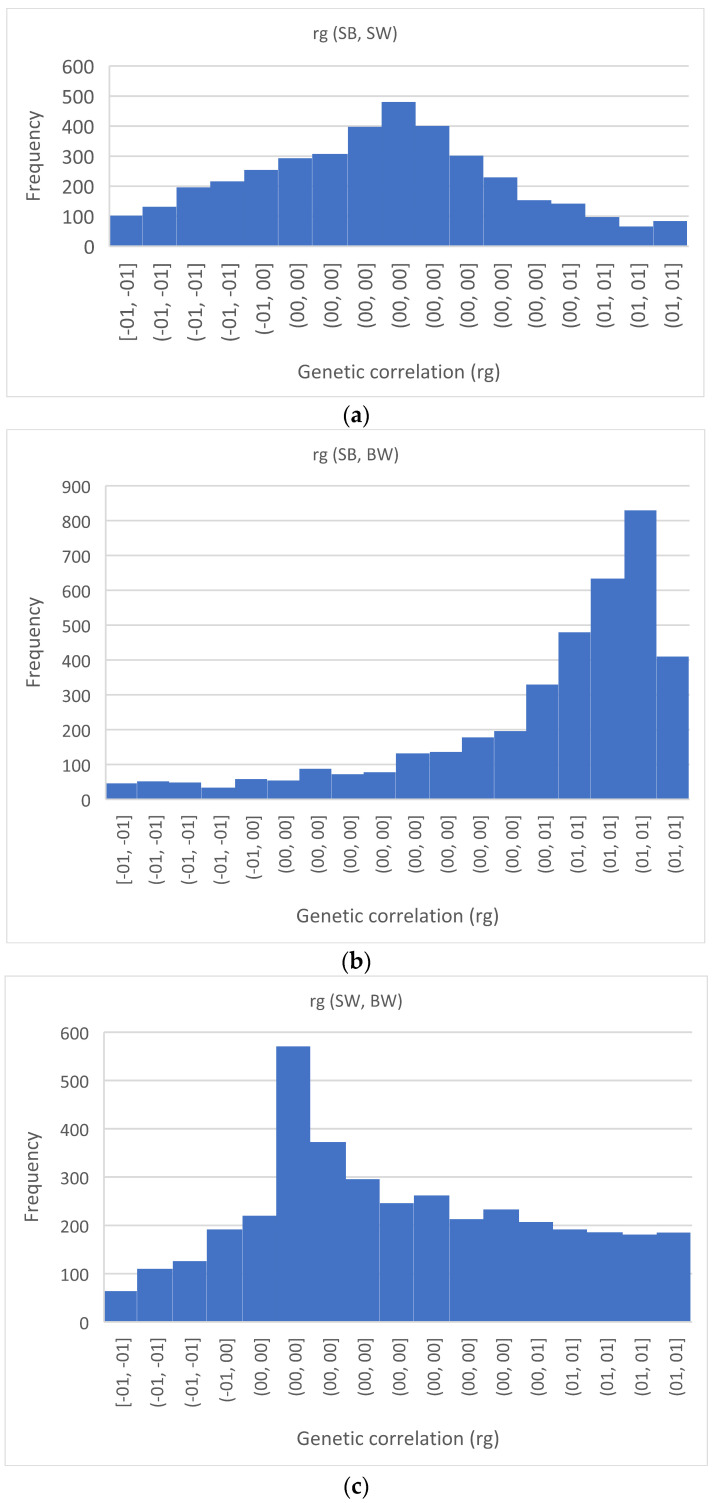
Marginal posterior distribution of genetic correlation (rg) between (**a**) survival at birth (SB) and weaning (SW), (**b**) survival at birth (SB) and individual birth weight (BW) and (**c**) survival at weaning (SW) and individual birth weight (BW).

**Table 1 animals-12-02695-t001:** Temperature and relative humidity by month.

		Temperature Outside (°C)		Temperature Inside (°C)		Relative Humidity Inside (%)	
		Minimun	Maximum	Minimun	Maximum	Minimun	Maximum
Summer	June	25	30	22	28	21	82
	July	27	39	26	33	21	80
	August	30	35	28	36	23	80
	Average	27.3	34.7	25.3	32.3	21.7	80.7
Fall	September	24	30	22	27	19	83
	October	21	27	20	23	21	79
	November	16	21	19	23	20	68
	Average	20.3	26.0	20.3	24.3	20.0	76.7

**Table 2 animals-12-02695-t002:** Number of observations (N), means and standard deviations (SD) of individual birth weight, litter size and kit survival traits.

Traits	N	Mean	SD
Individual birth weight (g) (with dead kits)	1696	51.36	15.19
Individual birth weight (g) (without dead kits)	1536	52.79	14.14
Individual birth weight of dead kits (g)	160	37.67	17.93
Number of kits born per litter	208	8.15	3.16
Number of kits born alive per litter	208	7.37	2.96
Number of kits weaned per litter	208	6.59	2.41
Survival at birth (%) ^a^	1536	90.56	23.03
Survival at weaning (%) ^b^	1373	89.41	24.47

^a^: Number kits born alive as proportion of total number kits born. ^b^: Number kits weaned as proportion of number of kits born alive.

**Table 3 animals-12-02695-t003:** Number of observations (N) and birth weight for kits alive and dead at birth and weaning.

Factors	Birth		Weaning	
	N	Weight (g)	N	Weight (g)
Parity				
First				
Live	617	46.84	467	48.1
Dead	61	35.33	150	42.9
Second				
Live	461	54.34	397	56.48
Dead	47	39.94	64	41.1
Third				
Live	458	59.25	395	61.37
Dead	52	37.21	63	45.97
Lactation status				
Lactation				
Live	881	49.8	690	51.51
Dead	80	35.8	191	43.58
Non-lactation				
Live	655	56.81	569	59.1
Dead	80	39.1	86	42.28
Season of farrowing				
Summer				
Live	945	49.22	753	58.81
Dead	97	37.45	192	42.97
Autumn				
Live	591	58.49	506	60.99
Dead	63	37.04	85	43.65
Nest quality				
Bad				
Live	20	52.5	17	51
Dead	3	51.33	3	61
Intermediate				
Live	79	52.45	62	54.43
Dead	10	47.9	17	45.23
Excellent				
Live	1437	52.84	1180	54.99
Dead	147	36.28	257	40.45
Cannibalism				
Non				
Live	1492	52.94	1222	55.1
Dead	145	37.99	270	43.2
Yes				
Live	44	47.52	37	48.54
Dead	15	30.53	7	42.14
Place of birth in the cage				
Inside nest				
Live	1467	52.49	1201	54.63
Dead	138	35.21	266	42.79
Outside nest				
Live	69	59.24	58	60.53
Dead	22	50.36	11	52.45
Sex				
Male				
Live	777	53.19	620	55.34
Dead	67	40	157	44.73
Female				
Live	759	52.38	639	54.48
Dead	93	35.34	120	41.15

**Table 4 animals-12-02695-t004:** Posterior means of heritabilities (diagonal), genetic correlations (above diagonal) and phenotypic correlations (below diagonal) for survival at birth (SB) and at weaning (SW), and individual birth weight (BW).

	SB	SW	BW
SB	0.018 [0.001, 0.052]	−0.111 [−0.763, 0.613] P (r_g_ < 0) = 0.67	+0.134 [−0.953, 0.996] P (r_g_ > 0) = 0.63
SW	−0.008 [−0.099, 0.089] P (r_p_ < 0) = 0.57	0.023 [0.002, 0.063]	+0.041 [−0.703, 0.935] P (r_g_ > 0) = 0.47
BW	+0.341 [0.260, 0.409] P (r_p_ > 0) = 1.00	0.248 [0.162, 0.328] P (r_p_ > 0) = 1.00	0.088 [0.006, 0.234]

Highest posterior density interval at 95% in brackets. P (r_g_ > 0): Probability of genetic correlation being higher than zero. P (r_g_ < 0): Probability of genetic correlation being lower than zero. P (r_p_ > 0): Probability of phenotypic correlation being higher than zero. P (r_p_ < 0): Probability of phenotypic correlation being lower than zero.

**Table 5 animals-12-02695-t005:** Posterior means of heritabilities (diagonal), genetic correlations (above diagonal) and phenotypic correlations (below diagonal) for survival at birth (SB) and at weaning (SW), and individual birth weight (BW) after adjustment for litter size.

	SB	SW	BW
SB	0.021 [0.001, 0.057]	−0.072 [−0.744, 0.654] P (rg < 0) = 0.56	+0.535 [−0.490, 0.944] P (rg > 0) = 0.87
SW	−0.009 [−0.099, 0.096] P (r_p_ < 0) = 0.58	0.027 [0.003, 0.071]	+0.083 [−0.628, 0.892] P (rg > 0) = 0.52
BW	+0.355 [0.290, 0.419] P (r_p_ > 0) = 1.00	+0.228 [0.155, 0.305] P (r_p_ > 0) = 1.00	0.146 [0.039, 0.292]

Highest posterior density interval at 95% in brackets. P (r_g_ > 0): Probability of genetic correlation being higher than zero. P (r_g_ < 0): Probability of genetic correlation being lower than zero. P (r_p_ > 0): Probability of phenotypic correlation being higher than zero. P (r_p_ < 0): Probability of phenotypic correlation being lower than zero.

**Table 6 animals-12-02695-t006:** Posterior means of phenotypic proportion of the maternal permanent effect (diagonal), and their correlations (above diagonal) for survival at birth (SB) and at weaning (SW), and individual birth weight (BW).

	SB	SW	BW
SB	0.043 [0.003, 0.106]	−0.003 [−0.791, 0.727] P (r_m_ < 0) = 0.50	−0.437 [−0.958, 0.335] P(r_m_ < 0) = 0.85
SW		0.024 [0.001, 0.071]	+0.098 [−0.923, 0.972] P(r_m_ > 0) = 0.57
BW			0.070 [0.007, 0,149]

Highest posterior density interval at 95% in brackets. P (r_m_ < 0): Probability of correlation of maternal effects being lower than zero. P (r_m_ > 0): Probability of correlation of maternal effects being larger than zero.

**Table 7 animals-12-02695-t007:** Posterior means of phenotypic proportion of the maternal permanent effect (diagonal), and their correlations (above diagonal) for survival at birth (SB) and at weaning (SW), and individual birth weight (BW) after adjustment for litter size.

	SB	SW	BW
SB	0.050 [0.005, 0.114]	+0.064 [−0.632, 0.862] P (r_m_ > 0) = 0.54	−0.702 [−0.996, −0.001] P (r_m_ < 0) = 0.95
SW		0.024 [0.001, 0.067]	−0.273 [−0.984, 0.657] P (r_m_ < 0) = 0.68
BW			0.037 [0.003, 0.093]

Highest posterior density interval at 95% in brackets. P (r_m_ < 0): Probability of correlation of maternal effects being lower than zero. P (r_m_ > 0): Probability of correlation of maternal effects being larger than zero.

**Table 8 animals-12-02695-t008:** Posterior means of phenotypic proportion of the litter variance (diagonal), and their correlations (above diagonal) for survival at birth (SB) and at weaning (SW), and individual birth weight (BW).

	SB	SW	BW
SB	0.199 [0.127, 0.270]	−0.029 [−0.435, 0.390] P (r_c_ < 0) = 0.55	+0.632 [0.384, 0.834] P (r_c_ > 0) = 0.99
SW		0.234 [0.161, 0.308]	+0.362 [0.111, 0.608] P (r_c_ > 0) = 0.99
BW			0.435 [0.351, 0.512]

Highest posterior density interval at 95% in brackets. P (r_c_ < 0): Probability of correlation of litter effects being lower than zero. P (r_c_ > 0): Probability of correlation of litter effects being larger than zero.

**Table 9 animals-12-02695-t009:** Posterior means of phenotypic proportion of the litter variance (diagonal), and their correlations (above diagonal) for survival at birth (SB) and at weaning (SW), and individual birth weight (BW) after adjustment for litter size.

	SB	SW	BW
SB	0.195 [0.125, 0.268]	−0.063 [−0.434, 0.454] P (r_c_ < 0) = 0.59	+0.768 [0.541, 0.962] P (r_c_ > 0) = 1.00
SW		0.225 [0.158, 0.298]	+0.312 [0.041, 0.596] P (r_c_ > 0) = 0.97
BW			0.251 [0.192, 0.312]

Highest posterior density interval at 95% in brackets. P (r_c_ < 0): Probability of correlation of litter effects being lower than zero. P (r_c_ > 0): Probability of correlation of litter effects being larger than zero.

## Data Availability

The data generated and analyzed during this study are included in this article.

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
