# Peer review of "Genetic Analyses of Rabbit Survival and Individual Birth Weight"

_animals, 2022, doi:10.3390/ani12192695_

Round 1
Reviewer 1 Report
The submitted article has an interesting subject focused on the estimates of genetic parameters for weight and individual survival in a specific rabbit line. Material, methods and results are well-structured. Some modifications are required.

Author Response
Author´s response
Manuscript number: Animals 1912895
Title: Genetic analyses of rabbit survival and individual birth weight
Dear Reviewer,
I would like to thank for their comments and suggestions.
We have modified the manuscript and all changes have been marked up using the "Track Changes" option.
. SIMPLE SUMMARY
L24: “and moderate (0.146) for birth weight after adjusted litter size”. This magnitude is still low. How did you consider that value as a moderate value?
We have rewritten the sentence to “Heritabilities of survivals at birth and weaning, as well as birth weight, were low (0.021 and 0.027) for survival traits and slightly greater (0.146) for birth weight after adjusted litter size
L27-28: That sentence refers to the magnitudes of genetic correlation “+0.134 and +0.535 after adjusted 26 for litter size” between the two traits. In that case, you could change “the magnitude of genetic parameters” to “these magnitudes of genetic correlations”
It was corrected.
. ABSTRACT
L42-43: The same comment as the previous one. Could you re-write?
It was corrected.
INTRODUCTION
General: The arguments are clear and justified.
L47: Replace “various” with “driver” or “factor”.
It was corrected.
L50: Replace “increasing of” with “increase in”.
It was replaced.
L52: Replace “under the influence of” with “influenced by”
It was replaced
L65 Are selection traits related to criteria or objectives of selection in the rabbit breeding program? Could you elaborate or add a word?
The sentence was rewritten.
L68-70: The objective was to estimate genetic parameters in birth weight too. As it is written, it seems that the objective was only to estimate heritability in survival traits, but not in individual birth weight. Add “individual”. Could you re-write this sentence, please?
The sentence was rewritten.
. MATERIALS AND METHODS
General: The structure is okay. Despite the justification of adjustment by litter size is known, you could add a brief sentence about the reason and implication to elucidate better here, in Materials and Methods.
It was included a sort sentence in line 136.
L97: Could you point out the precision of the scale used? (In the main text).
This information was included in the sentence “A total of 1696 kits from 208 litters of 81 does were weighted individually at birth using a digital scale up to 500 g with a precision of 0.01g”.
L76-97: Has not this population undergone some kind of selection?
ITELV2006 synthetic line has been maintained in discrete generation without selection and by avoiding inbreeding since its foundation. This information was included in M&M section of manuscript.
L105: So far, I cannot understand why the model did not include the maternal genetic effect for the traits. Could you clarify in that section?
Note that the effective separation of direct and maternal additive genetic effects requires large amounts of data, and our database is not large enough to obtain accurate estimates. This point could be particularly important for estimates of maternal genetic correlations between traits. Anyway, we tried to run a multivariate genetic analysis to estimate the direct and maternal additive genetic effects and maternal genetic correlations between traits, but we had serious problems and the model did not converge.
L112: Are there noticeable differences between the gender of kits in their early life for birth and survival traits? What was the base or kind of evaluation in order to include those fixed effects in the model?
The difference in birth weight ranged from 2% to 12% between males and females for lived and death kits at birth, this is reason to include the gender effect in the model.
L119: Could you improve the structure of the matrix, please?
We included additional information.
L125-126: Please, re-write the sentence.
The sentence was rewritten.
L139: Why did you use only Z criterion of Geweke as criterion of Convergence diagnostic? Effective sample size? Autocorrelation? Gelman-Rubin? Raftery-Lewis?
The sentence was rewritten and additional information was included.
L140: Remove “Bayesian confidence interval”. Confidence interval is related to frequentist statistics. The assumption is utterly different.
It was removed.
. RESULTS
General: I think you could put each table after the subsection related to it in the main text. That would make it easy and better reading.
It was made.
L155: Replace “including” with “with”.
It was replaced.
L155: Replace “excluding” with “without”.
It was replaced.
L100: “as high at 0.088”. That is not a high heritability. I suppose the idea of this sentence is to show the heritability estimate for individual birth weight is high (or less low) compared to survival traits. But, in summary, all estimates are very low. I disagree with this sentence.
The sentence was rewritten.
L173: What does “light” refer to? High, relevant, noticeable, slight?
The mistake was corrected.
L176: Replace “Except for” with “Apart from”.
It was replaced.
L186: Replace “Notice” with “Note”.
It was replaced.
L190: Replace “among” with “between”. The estimate of correlation used only two traits.
It was replaced.
L203-205: Could you re-write this sentence, please?
The sentence was rewritten.
L214: Replace “dam” with “maternal”. Please, you keep the same term when you refer to maternal permanent effect.
It was made.
L233: Replace “dam” with “maternal”.
It was made.
L245-250: I kindly recommend changing the quality of the figure. “rg” should be mentioned in the legend.
It was made.
. CONCLUSION
L233: Replace “will” with “would”.
L294-296: I disagree with this sentence. This research did not show any data of response to selection, then, how did you conclude that statement?
Conclusion section has been rewritten.

Reviewer 2 Report
The authors investigated the relation between birth weight and survivability in rabbit kits by means of Bayesian approach, and concluded that there is substantial potential for genetic improvement of kit survival at birth thought selection for birth weight.
I personally did not find any remarkable results that would lead to improved production efficiency of rabbit. The condition setting of the experiment is ambiguous, making it hard to determine whether the results are significant.
Line 82: “a building equipped with cooling system” does not provide any housing condition (temperature, humidity, etc.).
Line 91: Grading criteria of nest quality is unclear. What condition does “bad” indicate?
Line 150: What are the differences between summer and autumn (month, temperature, etc.)? Do seasonal differences affect housing condition in the building?
Other factors: Maternal and paternal conditions (age, weight, gestation period), etc.
Author Response
Author´s response
Manuscript number: Animals 1912895
Title: Genetic analyses of rabbit survival and individual birth weight
Dear Reviewer,
I would like to thank for your comments and suggestions.
We have modified the manuscript and all changes have been marked up using the "Track Changes" option.
The authors investigated the relation between birth weight and survivability in rabbit kits by means of Bayesian approach, and concluded that there is substantial potential for genetic improvement of kit survival at birth thought selection for birth weight.
I personally did not find any remarkable results that would lead to improved production efficiency of rabbit. The condition setting of the experiment is ambiguous, making it hard to determine whether the results are significant.
Conclusion section has been rewritten.
Line 82: “a building equipped with cooling system” does not provide any housing condition (temperature, humidity, etc.).
A table with temperatures and humidity was included in M&M section (Table 1).
Line 91: Grading criteria of nest quality is unclear. What condition does “bad” indicate?
We consider as bad nest when the female did not prepare her nest and the nest is empty without the presence of hair. We have added details about it in M&M section.
Line 150: What are the differences between summer and autumn (month, temperature, etc.)? Do seasonal differences affect housing condition in the building?
Table 1 shows the temperature and relative humidity by month, and we can see difference between summer and autumn.
Other factors: Maternal and paternal conditions (age, weight, gestation period), etc.
In relation to maternal conditions, these are included in the model through maternal (m) and common litter (c) effects. Regarding paternal effects, the model does not include them because the male was never in contact with the offspring.
Round 2
Reviewer 2 Report
The authors take the reviewers' suggestions seriously and have responded to them. However, there are still a few points that need to be revised.
Table 1 shows the temperature and relative humidity by month, and we can see difference between summer and autumn.
> What specific months are summer and fall respectively?
In relation to maternal conditions, these are included in the model through maternal (m) and common litter (c) effects. Regarding paternal effects, the model does not include them because the male was never in contact with the offspring.
>I think the conditions of the mated males (age, nutritional status or others) can affect the offspring (via sperm condition). Some explanation would be needed if the authors think that the effects of them can be eliminated.
Author Response
Author´s response
Manuscript number: Animals 1912895
Title: Genetic analyses of rabbit survival and individual birth weight
Dear Reviewer
Thank you for your comments. We have added in red colour the information that you required us in the new version of the manuscript.
. Comment 1. Table 1 shows the temperature and relative humidity by month, and we can see difference between summer and autumn.
What specific months are summer and fall respectively?
In M&M, we have added following sentence “Summer runs from June 1 to August 31, and fall from September 1 to November 30” in order to explain which months were included in summer and in fall, respectively. Moreover, Table 1 also includes this information.
. Comment 2. In relation to maternal conditions, these are included in the model through maternal (m) and common litter (c) effects. Regarding paternal effects, the model does not include them because the male was never in contact with the offspring.
I think the conditions of the mated males (age, nutritional status or others) can affect the offspring (via sperm condition). Some explanation would be needed if the authors think that the effects of them can be eliminated.
In M&M, we have added the following paragraph to clarify this point in the manuscript “Natural mating was performed. Adult males (7-10 months of age) were used. A maximum of two matings per week were carry out and the same feeding was used throughout the experiment, in order to guarantee qualitatively and quantitatively quality of semen”.
